# Efficiency of RAPD and SCoT Markers in the Genetic Diversity Assessment of the Common Bean

**DOI:** 10.3390/plants12152763

**Published:** 2023-07-25

**Authors:** Zuzana Hromadová, Zdenka Gálová, Lucia Mikolášová, Želmíra Balážová, Martin Vivodík, Milan Chňapek

**Affiliations:** Institute of Biotechnology, Faculty of Biotechnology and Food Sciences, Slovak University of Agriculture in Nitra, Tr. A. Hlinku 2, 949 76 Nitra, Slovakia; xhromadovaz@uniag.sk (Z.H.); zdenka.galova@uniag.sk (Z.G.); xmikolasova@uniag.sk (L.M.); zelmira.balazova@uniag.sk (Ž.B.); martin.vivodik@uniag.sk (M.V.)

**Keywords:** *Phaseolus vulgaris* L., molecular markers, DNA polymorphism, dendrogram, PCoA

## Abstract

Knowledge about the genetic diversity of the available common bean germplasm can help breeders properly direct the choice of genetic material in the breeding process. The aim of the present work was to estimate the usefulness of 10 RAPD and 10 SCoT markers in genetic diversity detection among 33 common bean genotypes. Both molecular marker systems were able to generate high levels of polymorphism in the genetic material, which was supported by the relatively high polymorphic information content (PIC) values observed for the used markers. The Diversity Detection Index (DDI) and Marker Index (MI) were used to compare the effectiveness of RAPD and SCoT markers. For both techniques, high values of MI and DDI were calculated, representing their effectivity. The SCoT markers showed higher values of the parameters used (MI = 7.474, DI = 2.265) than the RAPD markers (MI = 5.323, DDI = 1.612), indicating their higher efficiency in the detection of molecular variability. Three constructed dendrograms and PCoA plots were created using RAPD and SCoT, and both methods combined confirmed sufficient separation of the bean genotypes from each other. At the same time, a higher efficiency of SCoT markers compared to RAPD markers in the detection of the genetic diversity of beans was also proven. The results may be of future interest in the choice of genetically distant material for breeding purposes.

## 1. Introduction

The common bean (*Phaseolus vulgaris* L., 2n = 22) is a significant vegetable and grain legume of worldwide importance [1]. Since its domestication in Central and South America approximately 8000 years ago, the common bean has become one of the main legume crops with nutritional, economic, and environmental importance [2]. Beans are harvested and used as a vegetable in the form of fresh pods or as a grain legume in the form of dry or immature grains. Additionally, the leaves are sometimes used as vegetables [3].

With their high content of vitamins, minerals, carbohydrates, unsaturated fatty acids, and especially proteins, beans serve as a major source of these essential substances for poverty-stricken people in the countries of the developing world [4]. Up to 80% of beans are grown for food for the human population and animal feed [5]. However, it is estimated that in order to fulfil the food requirements and ensure nourishment for the rising world population, it will be necessary to increase world crop production by more than 60% by 2050 [6]. The introduction of new food crop cultivars with preferable nutritive quality, enhanced resistance to abiotic and biotic stresses, and improved yields is currently considered to be the best solution for the emerging problems [7]. In this context, many of the beneficial properties of legumes contribute to their perception as a key plant resource in sustainable food systems. The main qualities of legumes include the high content of nutritionally valuable proteins [8], the increase of soil fertility and minimised use of mineral fertilisers due to the ability to fix nitrogen [9], the lower production of greenhouse gases [10], and the improvement of plant biodiversity and soil physical conditions [11].

The common bean is a diploid (2n = 22) legume with 11 pairs of relatively small and morphologically similar chromosomes (around 2 µm) [12]. The genome size is 587 Mb, in which 27,197 protein-coding loci have been identified, including 4491 alternative transcripts [13]. Access to the sequences of common bean reference genomes, as well as the assumed functions of genes, offered new opportunities for marker-assisted selection and improvement of the effectiveness of common bean breeding [14]. The examination of genetic variability is a frequently used method for the characterization of genetic resources [15]. The collected information about the genetic diversity of common bean genotypes leads to efficient management of plant germplasm and enables breeders to use its full potential in breeding strategies [16]. For this purpose, various types of molecular markers were developed and applied to assess the genetic diversity among the genotypes [17]. In the different marker systems, numerous Polymerase Chain Reaction (PCR)-based markers were proven to be efficient for the investigation of genetic variability in collections of plant species [18], including the common bean. Many molecular markers, including RAPD [19,20,21], AFLP [22,23,24], ISSR [25,26], SSR [27,28,29], and SCoT [30,31], were successfully applied in the molecular variability analyses of the common bean. Random Amplified Polymorphic DNA (RAPD) markers are one of the types of arbitrarily amplified dominant markers. They are widely used in the research of plant population genetics, qualitative trait loci mapping, or DNA fingerprinting [32]. The use of a random primer enables us to conduct a polymorphism analysis without the need for prior knowledge about the DNA sequence of the analysed organism. This is one of the main advantages of the RAPD technique [33]. The genetic diversity of the common bean was detected with the use of RAPD [20]. They used 19 RAPD markers for the analysis of genetic relationships among 45 common bean genotypes. The authors of [19] successfully used the RAPD technique for the evaluation of genetic variability of French bean genotypes from different geographical regions of India. In their study [21], they focused on the agronomic, molecular, and nutritive characterisation of 13 kidney bean genotypes. For the molecular analysis, they used 10 RAPD markers to reveal the genetic relationships among the analysed bean genotypes. [23] studied the genetic structure in the collection of 200 common bean accessions from different parts of Mexico using the Amplified Fragment Length Polymorphism (AFLP) and Simple Sequence Repeats (SSR) markers. In their study, [24] were able to detect a high genetic variability in the collection of Brazilian carioca bean genotypes by the AFLP and SSR markers, suggesting their effectivity for the germplasm studies of bean. The authors of [22] analysed the genetic diversity of the common bean accessions using the AFLP markers and phaseolin seed protein. The Inter Simple Sequence Repeat (ISSR) markers were used by [25] in the investigation of the genetic diversity of the common bean. The authors applied seven ISSR markers to reveal the genetic structure of 12 Ethiopian bean accessions. The efficiency of ISSR markers in the genetic diversity studies of common beans was also proved by [26], who evaluated the genetic relationships among 57 common bean genotypes by means of 11 ISSR markers. [29] used the codominant Simple Sequence Repeats (SSR) markers to assess the genetic variability among 40 common bean landraces from Turkey. In their study, [28] reported on the effectivity of the SSR markers in the analysis of genetic relationships among the European common bean accessions. Contrary to the markers, which focus on the non-coding sequences of the genome, newer gene-targeted markers [34] have been developed in recent years, such as the Start Codon Targeted (SCoT) polymorphism. This marker system, developed by [35], is based on the short, conserved regions around the ATG start codon in a plant genome. The SCoT technique may be applied in the plant’s genetic variability and population relations studies as a standalone method or in combination with other molecular markers [36]. To our best knowledge, only two publications reporting on the use of SCoT markers in common bean genetic analyses are available to this day. The authors of [31] revealed a high genetic diversity in the set of 87 common bean accessions using eight SCoT markers. Based on the results, they considered using SCoT markers as a suitable technique for the genetic identification of genetic material based on its geographic origin. The sufficiency of SCoT markers in the genetic studies is also supported by the results presented by [30], who used five SCoT makers for the differentiation and genetic diversity study of 34 bean genotypes.

Each marker type has its advantages and disadvantages. The combination of several systems can be instrumental in getting a better understanding of the genetic diversity in the analysed species. The study of the genetic diversity of the common bean in the germplasm collections may enrich the existing knowledge of the genetic material stored in gene banks and direct its best use in breeding programmes. Therefore, the main aim of this study was to analyse the genetic variability in the collection of common bean genotypes by two different molecular marker systems (RAPD and SCoT) and to estimate their effectiveness in the genetic diversity studies of the common bean.

## 2. Results

For the genetic variability analysis, altogether 20 RAPD and SCoT markers were used to explore the level of DNA polymorphism in the collection of 33 common bean genotypes (Table 4). Both sets of DNA primers (10 RAPD and 10 SCoT listed in Table 5) produced distinct and polymorphic patterns in the genotypes.

### 2.1. RAPD Markers

Ten RAPD primers (Table 5) amplified a total of 96 PCR products. Out of these, 70 fragments (72.91%) were polymorphic, and 26 fragments (27.09%) were monomorphic (Table 1). The number of fragments produced by one primer varied from six (OPC-13; Appendix A) to fifteen (OPD-18; Figure 1), and the number of polymorphic fragments detected per primer varied from four (OPC-08 and OPC-13) to eleven (OPA-09 and OPD-18), with an average of seven polymorphic fragments per one primer. The average percentage of polymorphism in the bean collection observed by selected RAPD markers was 71.71%, with values of polymorphism varying from 50% (OPC-08) to 100% (OPC-04). The size of the products of the RAPD–PCR amplification ranged from 100 to 3000 bp. The values of polymorphic information content (PIC) in the collection of the RAPD markers used varied from 0.531 (OPD-19) to 0.955 (Sigma-D14), with an average value of 0.760.

The hierarchical cluster analysis based on the RAPD polymorphism (Figure 2) grouped 33 common bean genotypes into two main clusters (one and two), which were further separated into subclusters (1a, 1b, 2a, and 2b) based on their genetic similarity. Cluster one included nine bean genotypes subdivided into two subclusters (1a and 1b). In subcluster 1a, only the genotype Enso (21), originating from Sweden, was separated. Subcluster 1b contained eight genotypes of the common bean, which were further subdivided into two subgroups. In the first subgroup, the genotype Fruca Simpla (23) originating from Italy was separated from the others. The second subgroup was further divided into two subgroups containing three and four genotypes, respectively.

The second main cluster (two) separated 24 genotypes into two subclusters (2a, 2b). Subcluster 2a consisted of the genotype Goliat (8) originating from Poland. Subcluster 2b subdivided 23 genotypes into two subgroups of four and nineteen common bean genotypes, respectively. The genetic similarity according to the Jaccard coefficient varied from 0.494 to 0.875. Based on the RAPD polymorphism, the genotypes Canada (six) originating from Canada and Marika (nine) originating from the Czech Republic were the most genetically similar, with a value of Jaccard’s similarity coefficient of 0.875. On the other hand, the genotype Marika (nine) and genotype Atlanta (three) originating from The Netherlands were the most genetically distant. Therefore, these two genotypes are the most suitable candidates for marker-assisted breeding.

### 2.2. SCoT Markers

The ScoT primers amplified a slightly higher amount of scorable PCR products than the RAPD primers. Among the 117 amplified fragments, which were produced by 10 selected ScoT primers (Table 5), a total of 94 fragments (80.34%) were polymorphic, and 23 (19.66%) fragments were monomorphic (Table 2). The number of amplified fragments varied from seven (ScoT15) to seventeen (ScoT29; Figure 3), with an average of 11.7 fragments per marker. The highest number of polymorphic fragments (16) was detected by the SCoT29 marker, while the lowest number of polymorphic fragments (five) was observed using marker SCoT62. The SCoT polymorphism ranged from 55.56% (SCoT62) to 100% (SCoT15, SCoT30, and SCoT54; Appendix A), with an average of 80.71%. The size of fragments produced by SCoT-PCR varied from 180 to 3000 bp. The average value of polymorphic information content in the set of used SCoT markers was 0.795. The highest value of PIC (0.946) was detected using the SCoT33 marker. On the other hand, marker SCoT62 showed the lowest value of PIC (0.589).

Based on the SCoT data, the hierarchical cluster analysis separated common bean genotypes into two main clusters (one and two). Cluster one included genotype Enso (21) originating from Sweden, and cluster two grouped the remaining 32 genotypes (Figure 4), which were further subdivided into two subclusters (2a and 2b). Subcluster 2a included only genotype Marika (nine) from the Czech Republic. Subcluster 2b separated the genotype Start (33) originating from Hungary in one subgroup from other genotypes. The second subgroup of 2b included 29 common bean genotypes separated into subgroups according to their genetic similarity. The values of Jaccard’s similarity coefficient ranged from 0.439 to 0.909. Based on the SCoT polymorphism, the highest value of Jaccard’s similarity coefficient (0.909) was detected between genotypes Sancrop (14) from an unknown country of origin and Zlaty Roh (16) originating from the Slovak Republic. These two genotypes were genetically the closest. On the contrary, genotypes Sancrop (14) and Enso (21) originating from Sweden were the most genetically distant and can be recommended as a starting material for breeding the new lines.

The combined RAPD and SCoT molecular data were also used for the construction of a joint dendrogram. The hierarchical cluster analysis (Figure 5) grouped the common bean genotypes into two main clusters (one and two). Similarly, in the cluster analysis based on the SCoT polymorphism, cluster one separated the only genotype, Enso (21) originating from Sweden. Cluster two included the remaining 32 genotypes, which were subdivided into two subclusters (2a and 2b). Subcluster 2a included the genotype Start (33) originating from Hungary. Subcluster 2b separated the genotype Marika (nine), originating from the Czech Republic in one subgroup, from other genotypes. The second subgroup of 2b separated 29 common bean genotypes into groups of one and twenty-eight genotypes, respectively, according to their genetic similarity. The combined data revealed a lower range of Jaccard’s similarity coefficient values (0.494–0.855) in comparison with those of separate RAPD (0.494–0.875) and SCoT (0.439–0.909) data. Based on the combined RAPD and SCoT polymorphisms, the highest value of Jaccard’s similarity coefficient (0.855) was detected between genotype Sancrop (14), from an unknown country of origin, and genotype Zlaty Roh (16), from the Slovak Republic. This is in accordance with the results obtained from the SCoT polymorphism, where these two genotypes were considered genetically the closest based on Jaccard’s similarity coefficient (0.909). On the other hand, the highest genetic distance was detected between genotype Marika (nine), originating from the Czech Republic, and genotype Atlanta (three), originating from the Netherlands. The value of Jaccard’s similarity coefficient between these two genotypes was 0.494. A low value of Jaccard’s similarity coefficient (0.517) was also detected among genotype Enso (21) originating from Sweden and two other genotypes, Amethyst (17) originating from the Netherlands and Fruca Simpla (23) originating from Italy. Based on the obtained results, it can be concluded that 10 selected RAPD and 10 SCoT markers were polymorphic enough to distinguish 33 bean genotypes based on their genetic origin.

The data from the combined RAPD and SCoT binary matrix were also used to construct the three-dimensional plot based on the PCoA analysis (Figure 6). The results of PCoA used to determine the spatial representation of genetic distances among the bean varieties were consistent with the results of genetic differentiation based on cluster analysis (Figure 5). The PCoA plot grouped the bean genotypes into two main groups. The genotypes grouped in the red circle in the 1st and 2nd quadrants were separated from the genotypes in the green circle in the 3rd quadrant based on the joint results from both RAPD and SCoT methods (Figure 5). The results also showed that four varieties [Wawero (fifteen) originating from Germany, Start (thirty-three) originating from Hungary, Enso (twenty-one) originating from Sweden, and Marika (nine) originating from the Czech Republic] are genetically the most distant from the other analysed varieties. Therefore, the larger genetic distance of these four varieties may provide rich genetic resources to satisfy breeding requirements.

To compare the effectivity of the RAPD and SCoT markers used in the genetic diversity analysis of the common bean genotypes, different indicators were calculated (Table 3). RAPD-PCR analysis yielded a total of 96 fragments using 10 markers, while 10 SCoT markers amplified 117 fragments. On average, RAPD markers yielded 9.6 fragments per marker, while SCoT markers amplified an average of 11.7 fragments per marker. The average number of polymorphic fragments was 7 for the RAPD markers and 9.4 for the SCoT markers. Higher values of PIC (0.795), MI (7.474), and DDI (2.265) were detected by the application of SCoT markers, which indicates that these markers more effectively detect DNA polymorphism in bean genotypes compared to RAPD markers [PIC (0.760), MI (5.323), and DDI (1.612)].

## 3. Discussion

The study of genetic diversity is an integral part of the crop improvement process. The discovery of genetic relationships among plant genotypes based on DNA polymorphism may contribute to important knowledge leading not only to the simplification of the selection of germplasm for the breeding process but also to efficient conservation strategies [37,38] of genetic resources in gene banks. Many types of developed molecular markers have been used for the detection of genetic diversity in plants [30,31,39,40,41,42,43,44,45].

The individual marker techniques target various regions of the genome. The use of more than one marker system may provide a more informative approach while estimating the genetic diversity among the germplasm, as it might target different genomic regions. Therefore, the present study focuses on the comparison of the effectiveness of two molecular marker techniques (RAPD and SCoT) to detect genetic polymorphism in a set of 33 common bean genotypes. This information may serve as a significant means to access molecular variability and characterise the genetic relationships among the genotypes.

Each of the 20 markers used was able to produce different and polymorphic DNA profiles. The total number of amplified fragments per primer, together with PIC, MI, and EMR (effective multiplex ratio) [34], is regarded as one of the most significant indicators to assess the informativeness and efficiency of the markers used in the analyses of genetic diversity. In our study, RAPD-PCR analysis yielded on average 9.6 amplified fragments per marker using 10 markers, while 10 SCoT markers produced an average of 11.7 fragments per marker. The average number of polymorphic fragments was seven for the RAPD markers and 9.4 for the SCoT markers. Jose et al. [46] used the RAPD markers in a diversity analysis of the common bean landraces, reported a lower number (7.8) of amplified fragments per primer compared to our results. Despite the moderately high levels of polymorphism, with an average of 63.5%, they were able to reveal a wide genetic diversity among the selected common bean landraces.

Both sets of markers used in the study were able to generate considerable levels of polymorphism, indicating the suitability of these markers for the distinction of the genetic material. The RAPD markers revealed a lower average percentage of polymorphism (71.71%) than the SCoT markers (80.71%). High levels of polymorphism in the common bean germplasm, which are in accordance with our results, were earlier observed by several authors who used the RAPD and SCoT analyses [31,47,48,49]. Dursun et al. [47] were able to detect an average polymorphism level of 80% in the common bean breeding lines through the application of eight RAPD markers. Biswas et al. [48] successfully evaluated the genetic diversity among fourteen genotypes of French bean. They used six primers, which generated an average polymorphism of 70%. Tiwari et al. [49] studied the genetic variability of 99 bean genotypes using 10 RAPD, obtained an average of 12.3 fragments per primer, and found a DNA polymorphism of 91.06%. Eight SCoT primers were tested by [31] to analyse the genetic diversity of 87 bean genotypes. In their study, 14.75 fragments per primer were produced, and the average level of polymorphism was 87.51%.

The values of polymorphic information content represent the level of marker effectivity in the determination of polymorphism in the set of genotypes. The value of PIC depends not only on the number of produced fragments but also on their relative frequency [50,51]. On average, we detected higher average values of PIC in the SCoT markers (0.795) in comparison to the RAPD markers (0.760). In general, the high average values of PIC observed in both sets of markers indicate their high efficiency in the detection of polymorphism in the samples. In accordance with our results, the higher average values of PIC for the SCoT markers compared to the RAPD markers were also observed in works where the genetic diversity of some other *Fabaceae* species was detected, such as soybean [52] and field pea [53]. Osman and Ali [53] detected an average PIC value of 0.228 for SCoT markers, whereas for RAPD markers they observed an average PIC value of 0.180.

The suitability of RAPD and SCoT markers was compared by calculating two parameters, which are often used for the evaluation of the general effectiveness of marker systems in genetic diversity studies: the Marker Index (MI) and Diversity Detecting Index (DDI) (Table 3). Higher DDI (2.265) and MI (7.474) values were detected in the SCoT technique in comparison to lower DDI (1.612) and MI (5.323) values, which were detected in the RAPD markers. Currently, there have been no published works that compare the effectiveness of the SCoT and RAPD marker systems for the detection of the genetic diversity of common beans. In this respect, our work can be considered original for the common bean. Higher MI values for SCoT markers compared to MI values for RAPD markers were previously reported by [54,55] when analysing the genetic diversity of various food crops.

The comparison of some significant parameters (number of produced fragments per primer, percentage of polymorphic fragments, PIC, MI, and DDI), which are used for the evaluation of the marker system’s efficiency, revealed a slightly higher informative character of the SCoT markers compared to the RAPD markers. In general, based on the observed MI and DDI values as well as high polymorphism levels, we may consider the SCoT and RAPD markers to be effective tools suitable for the assessment of genetic variability in a collection of common beans. However, due to the detection of relatively low values of PIC with two of the used markers (SCoT62 and OPD-19), we would not recommend them for further analyses of common bean germplasm.

The study of molecular variability in crops based on DNA markers provides some essential information that is very useful for future crop improvement [37]. The main assumption for the success of breeding programmes is the proper selection of parental genotypes [56]. Hierarchical cluster analysis is a method commonly used for the depiction of relationships in the collection of genetic material.

In our study, a hierarchical cluster analysis based on the RAPD, SCoT, and combined data from the RAPD and SCoT markers was able to successfully distinguish among the genotypes and was proven to be a suitable approach for the evaluation of genetic variability in the common bean germplasm. A UPGMA dendrogram, which is based on the joint RAPD and SCoT data, grouped the analysed material into two main clusters. In accordance with the hierarchical cluster analysis based on the SCoT polymorphism, cluster one (one) included only one genotype, Enso from Sweden (twenty-one), and cluster two grouped thirty-two genotypes, which were divided into subclusters according to their genetic similarity.

The placement of individual genotypes in the different subclusters of the dendrogram and their mutual genetic distance may be a valuable source of knowledge for future breeding strategies. The right choice of genotypes for breeding purposes may lead to increased genetic diversity and the production of genetic material with superior traits. In current research, the UPGMA dendrogram and the PCoA plots confirmed a clear separation of the genotypes of common beans from each other using the RAPD, SCoT, and joint results of both methods.

An effective use of more than one marker technique in the analysis of genetic variability was previously noted in several studies of plants [34,53,57,58]. For example, Osman and Ali [53] reported on the efficient use of three marker techniques (RAPD, ISSR, and SCoT) in the study of genetic relationships among subspecies of field peas. The combination of RAPD, ISSR, and SCoT markers allowed the authors to detect high levels of genetic variability in the analysed samples and effectively reveal genetic relationships. The authors observed a higher informativeness of the SCoT markers compared to ISSR and RAPD. Based on the comparison of the presented genetic relationships among the analysed genetic material, the authors concluded that the results in the dendrogram generated by the SCoT polymorphism were in accordance with the results in the dendrogram obtained by the combined data.

Indeed, our results showed that we can effectively discriminate the genetic differences and phylogenetic relationships among common bean varieties based on SCoT and RAPD markers. Furthermore, these results help establish the theoretical foundation for selection, genetic preservation, and breeding of the common bean varieties.

## 4. Materials and Methods

### 4.1. Plant Material and Genomic DNA Isolation

The genotypes of the common bean used in the present study were taken from the collections of genetic resources of the Gene Bank of the Slovak Republic at the Research Institute of Plant Production in Piešťany, Slovak Republic, and the Gene Bank of the Czech Republic at the Research Institute of Plant Production in Prague-Ruzyně, Czech Republic. The seed samples were provided based on cooperation with the gene banks of the Slovak Republic and Czech Republic, with the aim of analysing the genetic relationships in the set of common bean genotypes. Information on the genetic background of the samples is not currently available. A total of 33 common bean genotypes (Table 4) from different countries of origin were used for the genetic variability studies. The Genomic DNA was extracted from 100 mg of fresh leaves collected from 14 day old seedlings of each common bean genotype using the GeneJET protocol for the isolation and purification of the plant DNA (Thermo Scientific™, Waltham, MA, USA). The bean genotypes were grown in a growth chamber on humus soil. The concentration of DNA was measured by BioDrop (Biochrom, Cambridge, UK), and the final concentration of DNA was adjusted to 50 ng/μL. All DNA samples were stored at −20 °C.

### 4.2. RAPD–PCR Amplification

The RAPD molecular marker analysis was performed using 10 RAPD markers (Table 5). The amplification of RAPD fragments was performed using decamer oligonucleotides primers. The polymerase chain reactions were performed in a final volume of 25 μL containing 1.25 μL of genomic DNA (100 ng), 12.5 μL of MasterMix (Promega, Madison, WI, USA), 10.25 μL of nuclease-free water, and 1 μL of primer (10 pmol). The PCR products were amplified in a programmed thermal cycler (Biometra, Göttingen, Germany) according to conditions described by [59], with an initial denaturation at 93.5 °C for 3 min, followed by 45 cycles of denaturation at 93.5 °C for 1 min, primer annealing at 36 °C for 2 min, extension at 72 °C for 3 min, and final extension at 72 °C for 7 min.

### 4.3. SCoT–PCR Amplification

Altogether, 10 SCoT markers (Table 5) designed according to [35,60] were used for the SCoT molecular marker analysis. The PCR amplifications were performed in a volume of 15 μL containing 1.5 μL of genomic DNA (100 ng), 7.5 μL of MasterMix (Promega, Madison, WI, USA), 4.5 μL of nuclease-free water, and 1.5 μL of primer (10 pmol). A thermal cycler (Biometra; Göttingen, Germany) was used for the amplification of SCoT fragments following the conditions according to [35], with an initial denaturation at 94 °C for 3 min, followed by 35 cycles of denaturation at 94 °C for 1 min, primer annealing at 50 °C for 1 min, extension at 72 °C for 2 min, and final extension at 72 °C for 5 min.

**Table 5 plants-12-02763-t005:** List of markers used in the RAPD and SCoT analyses.

RAPD Markers			SCoT Markers		
Marker	Sequence (5′-3′)	% GC	Marker	Sequence (5′-3′)	% GC
OPA-03 ^1^	AGTCAGCCAC	60	**SCoT2 ^5^**	CAACAATGGCTACCACCC	56
OPA-05 ^1^	AGGGGTCTTG	60	**SCoT15 ^5^**	ACGACATGGCGACCGCGA	67
OPA-09 ^1^	GGGTAACGCC	70	**SCoT18 ^5^**	ACCATGGCTACCACCGCC	67
OPC-04 ^2^	CCGCATCTAC	70	**SCoT19 ^5^**	ACCATGGCTACCACCGGC	67
OPC-08 ^1^	TGGACCGGTG	70	**SCoT29 ^5^**	CCATGGCTACCACCGGCC	72
OPC-13 ^3^	AAGCCTCGTC	60	**SCoT30 ^5^**	CCATGGCTACCACCGGCG	72
OPD-18 ^1^	GAGAGCCAAC	60	**SCoT34 ^5^**	ACCATGGCTACCACCGCA	61
OPD-19 ^3^	CTGGGGACTT	60	**SCoT54 ^6^**	ACAATGGCTACCACCAGC	56
OPE-01 ^1^	CCCAAGGTCC	70	**SCoT62 ^6^**	ACCATGGCTACCACGGAG	61
Sigma-D14 ^4^	TCTCGCTCCA	60	**SCoT63 ^6^**	ACCATGGCTACCACGGGC	67

Legend: ^1^—designed according to [40], ^2^—designed according to [39], ^3^—designed according to [61], ^4^—designed according to [62], ^5^—designed according to [35], and ^6^—designed according to [60].

### 4.4. Electrophoretic Separation of Fragments

The PCR products of RAPD and SCoT amplifications were separated with horizontal electrophoresis in 1.5% agarose gels in a 1× TBE buffer (Tris—borate—EDTA) containing 0.5 μg/mL ethidium bromide at a constant voltage of 100 V for approximately 1 h, according to [52]. The DNA fragments on the agarose gels were visualised under UV light and documented by the PhotoDoc-It^®^ (Ultra-Violet Products Ltd., Cambridge, UK) camera system.

### 4.5. Data Analysis

Amplified RAPD and SCoT fragments were scored as present (1) or absent (0) in the agarose gel, and each of them was treated as independent. The binary data were used to assess the level of polymorphism. The size of the DNA fragments was estimated by comparison with the Quick-Load^®^ Purple 2-Log (New England Biolabs Inc., Ipswich, MA, USA) 1 kb DNA ladder. The binary values were used to compute pair-wise similarity based on the Jaccard coefficient for the preparation of the similarity matrices. The hierarchical cluster analysis of genotypes was performed using the unweighted pair-group method with an arithmetic averages algorithm (UPGMA), which yielded the constructed dendrograms using the software iTOL [63] (https://itol.embl.de/upload.cgi) accessed on 10 May 2023, which is available online. The PCoA (Principal Coordinate Analysis) plot was constructed using the free statistical programme R Project, version 4.0.5. The PIC (polymorphic information content) was calculated according to [64] for each RAPD and SCoT marker to determine the levels of polymorphism and the usability of markers in the genetic diversity studies of the analysed set of common bean genetic material. To compare the efficiency of the RAPD and SCoT marker systems, the marker index (MI) and diversity detection index (DDI) were determined according to [65]. The marker index is used to describe the overall ability of a marker system to detect polymorphism. The diversity-detecting index is used to estimate the required number of marker loci.

The PIC values were calculated for each RAPD and SCoT primer according to the following formula [64]:$$PIC=1-\left(\sum_{i=1}^{n}p_{i}^{2}\right)-\sum_{i=1}^{n-1}\sum_{j=i+1}^{n}2p_{i}^{2}\cdot p_{j}^{2}$$
*p_i_* and *p_j_* are the frequencies of the *i*th and *j*th fragments in genotypes.

MI and DDI were determined according to the following formulas [65]:MI = ANPFG * PIC,
DDI = PIC * NPL/NAG,

ANPFG—average number of polymorphic fragments per genotype;NPL—number of polymorphic loci;NAG—number of analysed genotypes;PIC—polymorphic information content.

## 5. Conclusions

Based on the results of the present study, we conclude that the chosen RAPD and SCoT markers were an appropriate approach for the study of genetic diversity in the collection of 33 common bean genotypes. In general, the RAPD and SCoT markers used were able to reveal high levels of polymorphism in the analysed genetic material. The SCoT markers were proven to be more informative as they revealed on average a subtly higher level of polymorphism (80.71%) across the studied genotypes in comparison to the RAPD markers, which detected an average polymorphism of 71.71%. Some of the characteristics of the markers, such as MI (RAPD = 5.323 and SCoT = 7.474) and DDI (RAPD = 1.612 and SCoT = 2.265), were proof of the efficiency of the selected markers for the study of genetic variability. Two molecular markers (OPD-19 and SCoT62) achieved lower PIC values. We may expect their lower effectivity for the detection of polymorphism in the analysed collection. The cluster analysis based on the RAPD, SCoT, and combined data was able to sufficiently differentiate among the individual bean genotypes. The dendrograms constructed using the UPGMA algorithm appropriately illustrate the genetic relationships among genotypes, which may be used as an important source of information and be of value in breeding programmes with the aim of expanding the genetic variability of germplasm.

## Figures and Tables

**Figure 1 plants-12-02763-f001:**
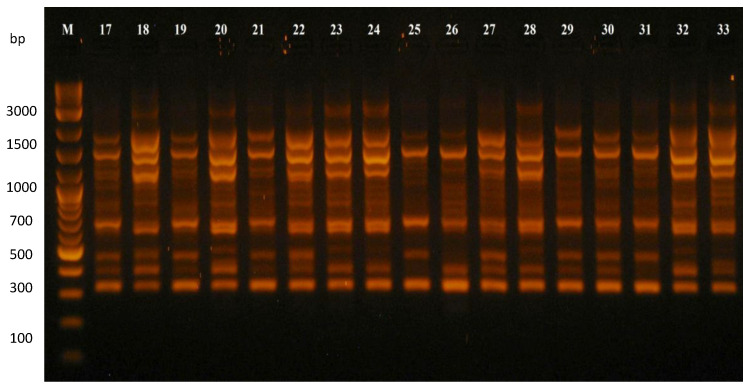
RAPD–PCR amplification products of the common bean genotypes generated by marker OPD-18. Notes: M—DNA ladder; 17–33—common bean genotypes listed in Table 4 under the numbers 17–33.

**Figure 2 plants-12-02763-f002:**
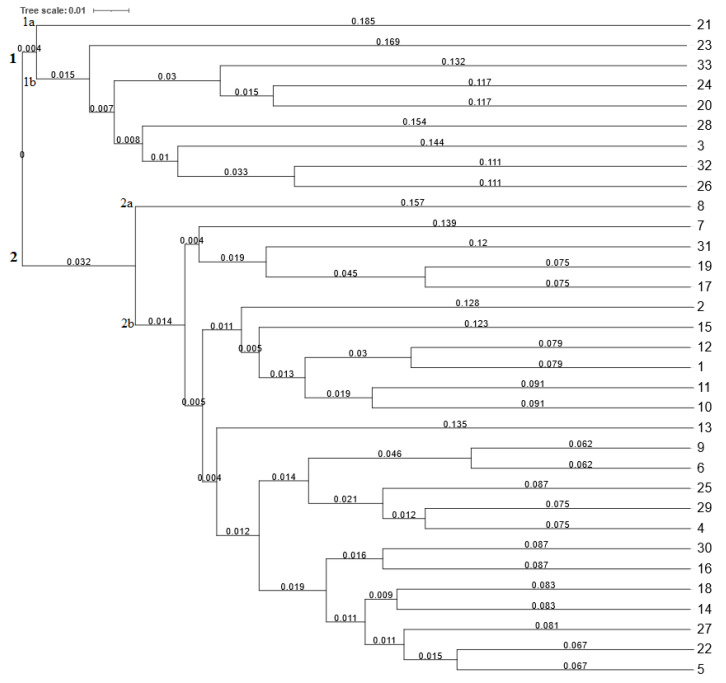
Dendrogram of 33 common bean genotypes constructed using 10 RAPD markers. Note: Branch lengths in a dendrogram generated by UPGMA cluster analysis represent the relative distances between individual branches of the dendrogram based on the similarity (Jaccard’s) coefficients. Numbers 1–33 correspond to the numbers of genotypes listed in Table 4.

**Figure 3 plants-12-02763-f003:**
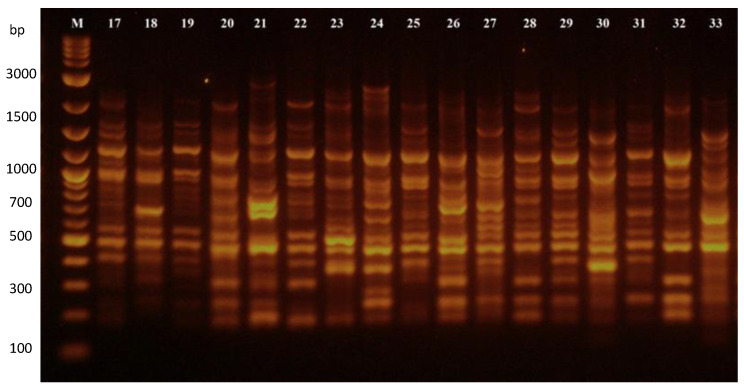
SCoT–PCR amplification products of common bean genotypes generated by marker SCoT29. Notes: M—DNA ladder; 17–33—common bean genotypes listed in Table 4 under the numbers 17–33.

**Figure 4 plants-12-02763-f004:**
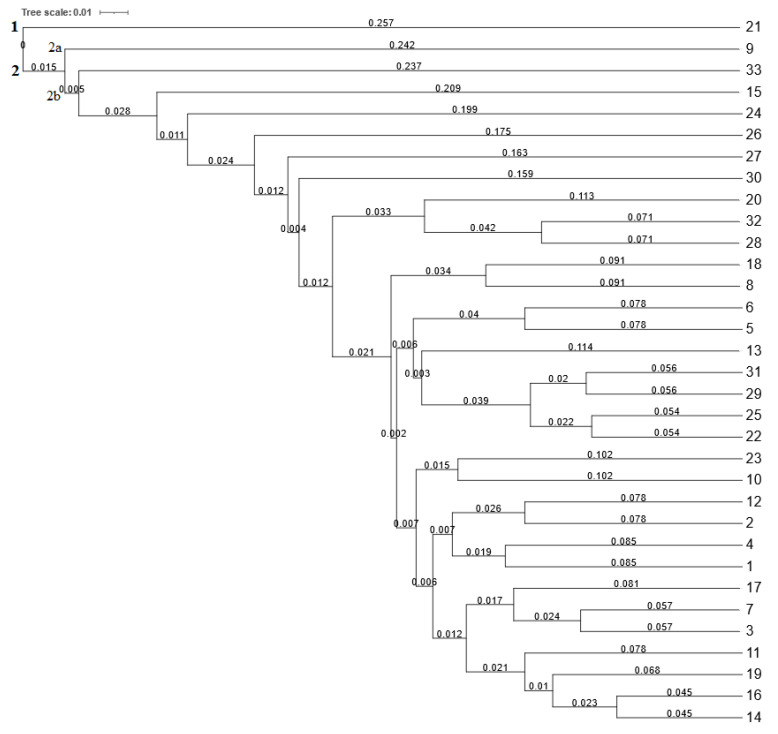
Dendrogram of 33 common bean genotypes constructed using 10 SCoT markers. Note: Branch lengths in a dendrogram generated by UPGMA cluster analysis represent the relative distances between individual branches of the dendrogram based on the similarity (Jaccard’s) coefficients. Numbers 1–33 correspond to the numbers of genotypes listed in Table 4.

**Figure 5 plants-12-02763-f005:**
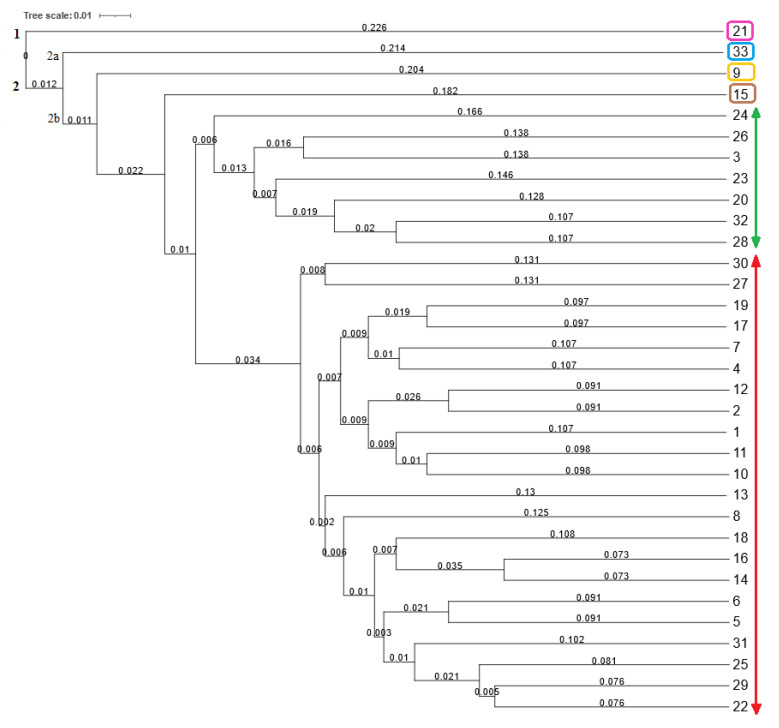
Joint dendrogram of 33 common bean genotypes constructed using 10 RAPD and 10 SCoT markers. Notes: branch lengths in a dendrogram generated by UPGMA cluster analysis represent the relative distances between individual branches of the dendrogram based on the similarity (Jaccard’s) coefficients. Numbers 1–33 correspond to the numbers of genotypes listed in Table 4. The red line corresponds to the red circle and the green line to the green circle in the PCoA plot (Figure 6). The pink, blue, yellow, and brown-coloured rectangles correspond to the separated genotypes in the circles of the same colours in the PCoA plot (Figure 6).

**Figure 6 plants-12-02763-f006:**
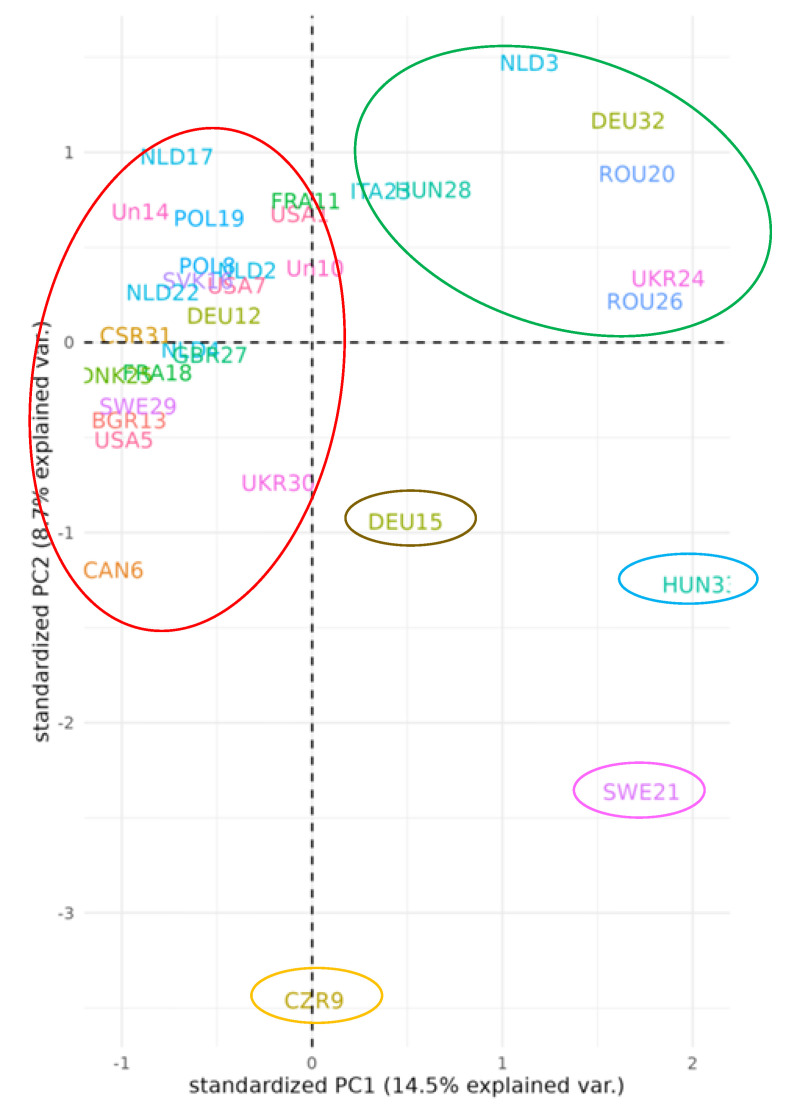
PCoA plot of 33 common bean genotypes based on the RAPD and SCoT markers. Notes: numbers 1–33 correspond to the numbers of genotypes listed in Table 4. The three—letter designation corresponds to the country of origin of the individual genotypes as follows: BGR—Bulgaria, CAN—Canada, CSR—Czechoslovakia, CZR—Czech Republic, DEU—Germany, DNK—Denmark, FRA—France, GBR—United Kingdom, HUN—Hungary, ITA—Italy, NLD—The Netherlands, POL—Poland, ROU—Romania, SVK—Slovak Republic, SWE—Sweden, UKR—Ukraine, un—unknown, and USA—United States of America. The red circle corresponds to the red line and the green circle to the green line in the cluster analysis (Figure 5). The pink, blue, yellow, and brown—coloured circles correspond to the genotypes marked by the rectangles of the same colours separated in the different clusters and subclusters in the cluster analysis (Figure 5).

**Table 1 plants-12-02763-t001:** Features of 10 RAPD markers.

Marker	TNF	NPF	PPF (%)	Molecular Weight Range (bp)	PIC
OPA-03	9	6	66.67	200–2000	0.677
OPA-05	10	6	60.00	400–2000	0.806
OPA-09	12	11	91.67	200–1500	0.839
OPC-04	9	9	100.00	600–1800	0.869
OPC-08	8	4	50.00	700–2000	0.733
OPC-13	6	4	66.67	400–3000	0.660
OPD-18	15	11	73.33	300–2000	0.872
OPD-19	9	5	55.56	400–3000	0.531
OPE-01	7	5	71.43	100–2000	0.663
Sigma-D14	11	9	81.82	280–2000	0.955
Average	9.6	7.0	71.71	-	0.760
Total	96	70	-	-	-
Range	-	-	-	100–3000	-

Legend: TNF—Total Number of Fragments, NPF—Number of Polymorphic Fragments, PPF—Percentage of Polymorphic Fragments, bp—base pair, and PIC—Polymorphic Information Content.

**Table 2 plants-12-02763-t002:** Features of the 10 SCoT markers.

Marker	TNF	NPF	PPF (%)	Molecular Weight Range (bp)	PIC
SCoT2	14	8	57.14	350–1900	0.780
SCoT15	7	7	100.00	200–800	0.740
SCoT19	14	9	64.29	900–2500	0.790
SCoT29	17	16	94.12	180–2800	0.919
SCoT30	14	14	100.00	250–2500	0.851
SCoT33	8	7	87.50	500–2500	0.946
SCoT34	12	8	66.67	230–1700	0.766
SCoT54	11	11	100.00	350–3000	0.825
SCoT62	9	5	55.56	380–2000	0.589
SCoT63	11	9	81.82	300–3000	0.744
Average	11.7	9.4	80.71	-	0.795
Total	117	94	-	-	-
Range	-	-	-	180–3000	-

Legend: TNF—Total Number of Fragments, NPF—Number of Polymorphic Fragments, PPF—Percentage of Polymorphic Fragments, bp—base pair, and PIC—Polymorphic Information Content.

**Table 3 plants-12-02763-t003:** Comparison of the efficiency of RAPD and SCoT markers for the genetic diversity assessment of the common bean.

Indicator	Type of Marker	
	RAPD	SCoT
Number of analysed genotypes	33	33
Number of primers	10	10
Total number of amplified fragments	96	117
Total number of polymorphic fragments	70	94
Number of polymorphic loci	70	94
Average number of polymorphic fragments per primer	7	9.4
Polymorphic information content	0.760	0.795
Polymorphic information content (range)	0.531–0.955	0.589–0.946
Marker index	5.323	7.474
Diversity detecting index	1.612	2.265

**Table 4 plants-12-02763-t004:** List of the 33 common bean (*Phaseolus vulgaris* L.) genotypes evaluated in the study.

Number	Genotype	Accession Code	Country of Origin
1.	Alicante ^a^	SVK001 L05 01130	USA
2.	Amanda ^a^	SVK001 L05 01032	The Netherlands
3.	Atlanta ^a^	SVK001 L05 01036	The Netherlands
4.	Belinda ^a^	SVK001 L05 01040	The Netherlands
5.	Cabernet ^a^	SVK001 L05 01131	USA
6.	Canada ^a^	SVK001 L05 01045	Canada
7.	Fullcrop ^a^	SVK001 L05 01065	USA
8.	Goliat ^a^	SVK001 L05 00455	Poland
9.	Marika ^a^	SVK001 L05 00932	Czech Republic
10.	Meteorit ^a^	SVK001 L05 01150	unknown
11.	Michael ^a^	SVK001 L05 01154	France
12.	Olga ^a^	SVK001 L05 00508	Germany
13.	Pesak ^a^	SVK001 L05 01166	Bulgaria
14.	Sancrop ^a^	SVK001 L05 01174	unknown
15.	Wawero ^a^	SVK001 L05 01188	Germany
16.	Zlaty Roh ^a^	SVK001 L05 01164	Slovak Republic
17.	Amethyst ^b^	09L0505134	The Netherlands
18.	Amulet ^b^	09L0505139	France
19.	Augustynka ^b^	05L0500062	Poland
20.	Avans ^b^	05L0500271	Romania
21.	Enso ^b^	09L0505322	Sweden
22.	Favorit ^b^	09L0505350	The Netherlands
23.	Fruca Simpla ^b^	09L0505384	Italy
24.	Gangtok bila ^b^	05L0500332	Ukraine
25.	Golden Dream ^b^	09L0505417	Denmark
26.	Grasa de Transilvania ^b^	09L0505437	Romania
27.	Herold ^b^	09L0505472	United Kingdom
28.	Kaboon ^b^	09L0500256	Hungary
29.	Katja ^b^	09L0500078	Sweden
30.	Kharkovskaya ^b^	05L0500151	Ukraine
31.	Mona ^b^	05L0500006	Czechoslovakia
32.	Nordstern ^b^	09L0500233	Germany
33.	Start ^b^	05L0500054	Hungary

Legend: ^a^—genotypes from the GeneBank of the Research Institute of Plant Production in Piešťany (Slovak Republic); and ^b^—genotypes from the GenBank of the Research Institute of Plant Production in Prague-Ruzyně (Czech Republic).

## Data Availability

The data presented in this study are available from the corresponding author upon reasonable request.

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
