# Peer review of "Efficiency of RAPD and SCoT Markers in the Genetic Diversity Assessment of the Common Bean"

_plants, 2023, doi:10.3390/plants12152763_

Round 1

Reviewer 1 Report

The manuscript is prepared on an interesting topic, where I deal with the application of two methods of studying genetic diversity to a collection of beans. However, the manuscript contains fundamental flaws that must be eliminated before its acceptance. Abstract and keywords - some keywords match the title of the manuscript. Introduction - some information in the introduction is repeated in the discussion section (e.g. line 261-263). Material and methodology - there is no information for what purpose the collection of varieties for analysis was created. This fact is essential because the choice of material (origin, etc.) can significantly affect the results achieved. The results - are logically ordered, but there is a fundamental deficiency within the dendrograms (Fig. 2, 4, and 5), as the significance of individual nodal points is not indicated (e.g. bootstrapping value). In their given form, all three dendrograms lack essential explanatory power. Discussion - authors try to discuss their results with other authors using publications on beans, but they also branch out to other crops. The question is whether in this case it is appropriate to use links to one's own publications (Yes, I understand the authors' effort to document their experience, but it is also appropriate to add the work of other teams/workplaces.). It would certainly be more appropriate to document that the given methods are also used in places other than the home workplace. In the discussion and comparison of the results, it would be appropriate to think about the comparison of self-pollinated, cross-pollinated and vegetative propagated plants. The method of reproduction can play a major role in the values ​​achieved. Figure 6 - can the differences in origin be supplemented by some differences in fundamental morphological characters? do the authors have this information? References - not all are processed according to the guidelines for authors. Especially slight confusions are in the format of journals, or the format and their standard forms of names are not unified. I recommend the authors to show at least one selected RAPD and SCoT marker for all analyzed genotypes (varieties) in the Supplements.

At the same time as editing the manuscript, I recommend a small proofreading of the English language to remove some less than ideal wording. Sometimes it is better to write shorter sentences than complicated sentences.

Author Response

We would like to thank you for reading our manuscript and writing the review. Here we respond to your comments and suggestions.

Keywords: Accepted.

Introduction: Accepted. The duplicate information present in the introduction and discussion were reduced (primarily in the discussion).

Materials and methods: The seed samples were provided on the basis of cooperation with the gene banks of the Slovak Republic and the Czech Republic for the purpose of DNA analysis and determination of genetic relationships between varieties. This information is missing in gene banks. Information on the genetic background of the samples is not currently available in gene banks. 

Results: Accepted. The lengths of individual branches were added to the dendrograms (Fig. 2, 4, and 5). Discussion – Accepted.  The discussion has been revised and supplemented.

Figure 6: (PCoA) “Can be the differences in the origin supported by some differences in the fundamental morphological characters? Do the authors have this information?”

Response: In this article, we did not evaluate the morphological characters of the seeds in relation to the genetic diversity of the samples. References: Accepted. References were processed according to the guidelines for authors. According to the manual, it is possible to leave the full journal title if we were not sure how to abbreviate a particular journal title.

 Supplements: Accepted. The Supplementary file shows electrophoretic profiles for all analyzed genotypes (Figure S1 and S2) for one RAPD marker (OPC-13) and one SCoT marker (SCoT54).

 Quality of English Language: The English language has been corrected. We edited sentences that were too long and complicated to write.

Reviewer 2 Report

The authors focused on the possibilities of evaluating the genetic diversity of common bean (33 genotypes) using 10 RAPD and 10 ScoT markers. The authors concluded that these sets of markers were sufficient to distinguish 33 bean genotypes based on their genetic origin. Based on these analyses, the authors pointed out some genotypes as a starting material for breeding the new lines. The data is interesting and worthy of publication. However, I have some comments or suggestions.

The authors state that the ScoT markers were more informative. Do the authors assume that ScoT markers can therefore be sufficient to evaluate the genetic diversity of the studied genotypes? How do they then explain that using RAPD markers they evaluated the genotypes Marika and Atlanta as the most genetically distant, while using ScoT markers they evaluated the genotypes Sancrop and Enso? Moreover, using combined RAPD and ScoT data they evaluated the genotypes with the highest genetic distance i) Marika and Atlanta and ii) Enso and 2 genotypes Amethyst and Fruca Simpla. To what extent are these data usable for assisted breeding?

The Discussion is too broad and needs some modification.

Page 10, line 285: Please check the construction of the sentence.

Page 13, line 416: I suggest renaming as Plant material and genomic DNA isolation.

Page 14, lines 432-440: The origin of the used RAPD primer sequences is not given, please add a citation or describe it in more details.

Author Response

We would like to thank you for reading our manuscript and writing the review.

The authors state that the ScoT markers were more informative. Do the authors assume that ScoT markers can therefore be sufficient to evaluate the genetic diversity of the studied genotypes?

Response: We assume that the higher informativeness of SCoT markers is mainly related to the fact that, unlike RAPD markers, which are designed randomly and target a non-coding region of the genome, SCoT markers are gene-specific. This statement is confirmed by Rai (2023), who states that SCoT markers are more useful than arbitrary markers like RAPD or ISSR, because it is derived from the gene itself or the flanking regions around it and produces markers linked with a specific trait. The use of a higher number of SCoT markers could provide a sufficient information of the genetic diversity of the analyzed genotypes.

Rai, M. K. (2023). Start codon targeted (SCoT) polymorphism marker in plant genome analysis: current status and prospects. Planta, 257(2). https://doi.org/10.1007/s00425-023-04067-6

How do they then explain that using RAPD markers they evaluated the genotypes Marika and Atlanta as the most genetically distant, while using ScoT markers they evaluated the genotypes Sancrop and Enso? Moreover, using combined RAPD and ScoT data they evaluated the genotypes with the highest genetic distance i) Marika and Atlanta and ii) Enso and 2 genotypes Amethyst and Fruca Simpla. To what extent are these data usable for assisted breeding?

Response: Differences in the representation of genetic relationships within individual dendrograms are due to the fact that each marker system targets different genomic sequences. ScoT markers bind to gene-specific sequences, while RAPD markers bind to non-coding sequences of the genome, which generates differences in genotype polymorphism profiles. The combination of data from both techniques provides a more comprehensive view of the genetic relationships of individual varieties, which is important for breeders in marker-assisted breeding. Discussion: Accepted. The discussion has been edited.

Page 10, line 285: Please check the construction of the sentence. à Accepted. The sentence has been completely removed.

Page 13, line 416: I suggest renaming as Plant material and genomic DNA isolation. à Accepted. The name was changed according to your suggestion.

Page 14, lines 432-440: The origin of the used RAPD primer sequences is not given, please add a citation or describe it in more details. à The paper citation for each of the RAPD primer sequences used is given in the legend below Table 5.

Round 2

Reviewer 1 Report

The authors handled all text edits in the manuscript. Nevertheless, I have a major comment about the modification of the similarity dendrograms, where the authors inserted the Jaccard similarity coefficients (without adding a legend to the individual dendrograms, so that it is self-explanatory). I see the processing as problematic. By default, the value of "bootstrapping" is given in case of nodal points (branching points) or the part of the dendrogram is usually a segment characterizing the similarity. Although this method is interesting, it does not contribute to the understanding of the similarity without further comment. I consider it appropriate to adhere to the standards for the presentation of dendrograms.

Author Response

Results: Accepted. Dendrograms (Fig. 2, Fig. 4, Fig. 5) have been edited. The text in data analysis was corrected. 

Reviewer 2 Report

The authors sufficiently answered my questions and made the required adjustments in the text. I have no further comments.

Author Response

The reviewer 2 did not require further modification of the manuscript.

Round 3

Reviewer 1 Report

The authors added "tree scale" and the legend to the dendrograms, but the information about the numerical values ​​is missing in the legend, i.e. that these are the values ​​of the similarity (Jaccard's) coefficients. Really, the images should be self-explanatory. I also have some small comments about the text of a formal nature: line 449 - there is a ")" after the website link, but where is the "(", which would delimit this section? I recommend that a careful check be made of the References section and the names of journals and proceedings that do not meet the requirements of the journal, i.e. line 506, 523, 660-661 vs. 665, 674.

These comments are only of a formal nature and after their removal the manuscript can be accepted for publication.

Author Response

The authors added "tree scale" and the legend to the dendrograms, but the information about the numerical values ​​is missing in the legend, i.e. that these are the values ​​of the similarity (Jaccard's) coefficients. Really, the images should be self-explanatory.

Accepted. Dendrograms (Fig. 2, Fig. 4, Fig. 5) have been edited. 

I also have some small comments about the text of a formal nature: line 449 - there is a ")" after the website link, but where is the "(", which would delimit this section? I recommend that a careful check be made of the References section and the names of journals and proceedings that do not meet the requirements of the journal, i.e. line 506, 523, 660-661 vs. 665, 674.

Accepted. References have been corrected.